# Treatment Sequencing in Chronic Lymphocytic Leukemia in 2024: Where We Are and Where We Are Headed

**DOI:** 10.3390/cancers16112011

**Published:** 2024-05-25

**Authors:** Alberto Fresa, Idanna Innocenti, Annamaria Tomasso, Luca Stirparo, Antonio Mosca, Francesco Iadevaia, Francesco Autore, Paolo Ghia, Luca Laurenti

**Affiliations:** 1Dipartimento di Diagnostica per Immagini, Radioterapia Oncologica ed Ematologia, Fondazione Policlinico Universitario Agostino Gemelli IRCCS, 00168 Rome, Italy; alberto.fresa92@gmail.com (A.F.); idanna.innocenti@policlinicogemelli.it (I.I.); francesco.autore@policlinicogemelli.it (F.A.); 2Sezione di Ematologia, Dipartimento di Scienze Radiologiche ed Ematologiche, Università Cattolica del Sacro Cuore, 00168 Rome, Italy; annamaria.tomasso01@icatt.it (A.T.); luca.stirparo01@icatt.it (L.S.); antonio.mosca03@icatt.it (A.M.); francesco.iadevaia01@icatt.it (F.I.); 3Division of Experimental Oncology, Università Vita-Salute San Raffaele, IRCCS Ospedale San Raffaele, 20132 Milan, Italy; ghia.paolo@hsr.it

**Keywords:** CLL, personalized, treatment, ibrutinib, acalabrutinib, zanubrutinib, venetoclax

## Abstract

**Simple Summary:**

The treatment of chronic lymphocytic leukemia is constantly evolving. Within the past few years, the treatment algorithm of patients with CLL has been radically transformed by the introduction of BTK and BCL2 inhibitors. This change has raised the need to optimize their use to ensure the greatest possible efficacy by personalizing therapies as much as possible. This review aims to clarify how to tailor the sequence of different treatments in CLL, and illustrate what resources are currently available to patients previously treated with the different molecules.

**Abstract:**

As treatments with BTK inhibitors and BCL2 inhibitors have replaced the use of chemoimmunotherapy in CLL in both first-line and relapsed patients, it becomes critical to rationalize their use and exploit the full potential of each drug. Despite their proven, robust, and manifest efficacy, BTKis and BCL2is fail to provide long-term disease control in some categories of patients, and to date this is an unmet clinical need that is critical to recognize and address. Ongoing clinical trials are evaluating new treatment algorithms and new molecules to progressively thin this population. In this review for each category of patients we explicate the different possible patterns of treatment sequencing based on currently available evidence, starting from the frontline to currently ongoing trials, in order to optimize therapies as much as possible.

## 1. Introduction

Chronic lymphocytic leukemia/small lymphocytic lymphoma (CLL/SLL) is the most common lymphoproliferative disorder in adults [1]. CLL is defined, in accordance with the World Health Organization and International Consensus Classification Clinical Advisory Committee’s 2022 Classification of Lymphoid Neoplasms updates, as a low-grade lymphoproliferative neoplasm with ≥5 × 10^9^/L clonal B cells in the peripheral circulation expressing CD5, CD19, CD20(dim), and CD23 [2,3].

Most CLL patients present with asymptomatic disease at the time of diagnosis. Only patients who meet the 2018 iwCLL criteria [4] for the initiation of treatment are referred for treatment. There are many predictive and prognostic markers in CLL, but a comprehensive review of these is beyond the scope of this management algorithm [5]. In clinical practice, one of the most effective scores for predicting time to first treatment (TTFT) is the CLL International Prognostic Index (CLL-IPI) [6,7]. It is calculated at the time of diagnosis and includes the following variables: TP53 status, IGHV mutational status, serum β2-microglobulin concentration, clinical stage, and age. Risk stratification can help in planning patients’ follow-up.

When treatment becomes necessary, the choice of frontline therapy depends on various factors, including patient age, fitness, and molecular and cytogenetic markers. The patients who meet the 2018 iwCLL criteria (Table 1) should be treated, independently of their CLL-IPI risk [4].

Given the consistent observations of improved progression-free survival (PFS) and overall survival (OS) with the use of Bruton’s tyrosine kinase inhibitors (BTKi) and B-cell lymphoma-2 inhibitor (BCL2i) compared with chemoimmunotherapy, the use of FCR (fludarabine, cyclophosphamide, and rituximab), BR (bendamustine and rituximab) or chlorambucil +/− anti-CD20 in the first-line treatment of CLL has progressively declined [8,9,10,11]. Currently, chemoimmunotherapy may be considered as an appropriate treatment option (FCR for patients < 65 years, BR for patients ≥ 65 years) for patients who have mutated IGHV genes and TP53 wild type, and where novel agents are not readily available [12]. Patient and disease characteristics, such as cytogenetics, fitness status, comorbidities, and patient preference, play a very important role in the choice of therapy, where some patients prefer fixed-duration combination therapy based on BCL2is over continuous oral therapy with BTKis. Outside clinical trials, our current approach to the first-line treatment of CLL patients who meet the 2018 iwCLL criteria for therapy is shown in Figure 1.

TP53 status is one of the most important prognostic and predictive biomarkers in CLL. It is critical to obtain both a FISH panel for del17p13 and an evaluation for TP53 mutations, as 3–5% of patients harbor a TP53 mutation on DNA sequencing in the absence of del17p13 at FISH, and these patients have poor outcomes as well [13,14,15,16,17,18,19,20].

## 2. Frontline Treatment

### 2.1. Del17p/TP53 Disruption Present

#### 2.1.1. Covalent BTKi +/− Anti-CD20

Chemoimmunotherapy in patients with TP53 alteration leads to inauspicious outcomes, with short PFS and OS [21,22]. In contrast, far better outcomes have been achieved by BTKis. A phase two trial testing ibrutinib monotherapy on TP53-altered patients on first-line treatment (n = 34) showed a 61% PFS at 6 years and a 79% OS at 6 years [23]. Superimposable results were reported from a pooled analysis of four clinical trials in which ibrutinib ± rituximab was used as a frontline treatment in TP53-disrupted patients (n = 89); 4-year PFS was 79% and 4-year OS was 88% [20]. In the ELEVATE-TN trial, acalabrutinib +/− obinutuzumab showed a benefit in PFS in high-risk genomic subgroups. In patients with del(17)(p13.1) and/or mutated TP53 at a median follow-up of 74.5 months, median PFS was 73.1 months for acalabrutinib–obinutuzumab and not reached for acalabrutinib monotherapy versus 17.5 months for obinutuzumab–chlorambucil (HR: 0.28 and 0.23, respectively; *p* ≤ 0.0009 for both); estimated 6-year PFS was 56% and 56% versus 18%; similar results were seen in patients with only del(17)(p13.1) [9,24]. In arm C of the SEQUOIA trial, zanubrutinib in patients with del(17)(p13.1) did not reach median PFS and OS. The estimated 18-month PFS rate was 88.6% (95% CI: 79.0–94.0), while the estimated 18-month OS rate was 95.1% (95% CI: 88.4–98.0) [25]. No consistent benefit was demonstrated with the addition of anti-CD20 antibodies (rituximab and obinutuzumab) to BTKi monotherapy in patients with TP53 alteration in the first-line management of CLL [9,24,26]. 

#### 2.1.2. Venetoclax +/− Anti-CD20

Fixed-duration venetoclax–obinutuzumab (VO) combination therapy demonstrated efficacy in the CLL14 trial [11]. In patients with TP53 disruption treated with VO, the median PFS was 51.9 months [27], suggesting that continuous BTKi-based treatment in patients with TP53 alteration may provide superior PFS, even if nowadays we lack studies comparing fixed and continuous therapies with long-term follow-up. The ongoing CLL17 study directly compares continuous BTKi and fixed-duration VO in treatment-naïve CLL (NCT04608318). Data suggest that there may also be significant differences in efficacy among other specific CLL subgroups. For example, the 5-year PFS across trials differs numerically in the unmutated IGHV subgroup treated with ibrutinib (67%) [28] compared with VO (56%) [27]. The difference in PFS is greater for patients with TP53 aberration, favoring continuous treatment with ibrutinib (70% at 5 years) [23] compared with VO (41% at 5 years) [27]. Despite the limited number of patients and considering the low statistical rigor of trial-to-trial comparison, however, an important question arises: whether a prolonged duration of therapy and/or BTK inhibition give a therapeutic advantage in patients harboring TP53 aberration in particular and, to a lesser extent, U-IGHV. The main results of the first-line trials are listed in Table 2.

### 2.2. Del17p/TP53 Disruption Absent

#### 2.2.1. Covalent BTKi +/− Anti-CD20

First-line ibrutinib was compared with chlorambucil in the RESONATE-2 study in elderly CLL patients without del17p. In this study, ibrutinib showed a superior outcome to chlorambucil, both in terms of PFS at 7 years (59% vs. 9%, respectively, *p* < 0.0001) and OS (median OS not reached vs. 89 months, respectively, with crossover allowed from chlorambucil to ibrutinib at disease progression) [28]. In the Alliance A041202547 study, which compared patients with CLL ≥ 65 years of age who received frontline ibrutinib, ibrutinib-rituximab, or BR, after a median follow-up of 55 months, the ibrutinib-containing arms showed significantly better PFS (76% at 4 years) than BR (47% at 4 years, *p* < 0.001), although no difference in OS was observed [29]. However, no difference was observed between the two ibrutinib-containing arms, suggesting no benefit in adding anti-CD20 to ibrutinib. The E1912 study, which compared frontline FCR and ibrutinib–rituximab in 529 younger (<70 years) CLL patients after a median follow-up of 5.8 years, showed superior PFS (5-year PFS: 78% vs. 51%, *p* < 0.0001) and OS (5-year OS: 95% vs. 89%, *p* = 0.02) in favor of ibrutinib–rituximab [30]. Ibrutinib–rituximab showed superior PFS in both IGHV-unmutated (HR 0.27, 5-year PFS 75% vs. 33%, *p* = 0.001) and IGHV-mutated (HR 0.27, 5-year PFS 83% vs. 63%, *p* = 0.001) groups. Finally, the FLAIR trial also demonstrated the superiority of frontline ibrutinib–rituximab over FCR in 771 CLL patients (median PFS not reached for ibrutinib–rituximab compared with 67 months for FCR; HR: 0.44; *p* < 0.001), but did not show better OS nor a PFS advantage in IGHV-mutated patients [31].

The other two major covalent BTK inhibitors, acalabrutinib and zanubrutinib, which are more selective for BTK, were studied in first-line CLL therapy. The ELEVATE-TN trial randomized 535 patients with CLL to receive first-line monotherapy with acalabrutinib, acalabrutinib–obinutuzumab, and chlorambucil–obinutuzumab (median age 70 years). After a median follow-up of 58.2 months, the PFS of acalabrutinib-containing regimens was superior to chlorambucil–obinutuzumab, with an estimated 5-year PFS of 84% with acalabrutinib–obinutuzumab versus 72% with acalabrutinib monotherapy versus 21% with chlorambucil–obinutuzumab (*p* < 0.0001) [32]. In that study, the combination of acalabrutinib with the second-generation anti-CD20 obinutuzumab would appear to confer a benefit in terms of PFS (HR 0.56, *p* = 0.0296), but the study was not powered to compare acalabrutinib with acalabrutinib–obinutuzumab, and more grade ≥ 3 neutropenia and grade ≥ 3 infections were reported in the anti-CD20 combination arm.

There was no difference in outcomes between the single-agent ibrutinib and ibrutinib–rituximab arms in multiple studies; hence, we generally do not use rituximab with BTKis [26,29]. However, given the PFS benefit observed in ELEVATE-TN, obinutuzumab can be used in combination with acalabrutinib, particularly for patients without TP53 alteration, although the higher incidence of complications should be taken into account [32].

In the SEQUOIA trial, 590 patients with CLL del17p-negative CLL were randomized to receive frontline zanubrutinib or BR (median age 70 years), and after a median follow-up of 26 months, the estimated PFS at 24 months was 85% with zanubrutinib compared with 69% with BR (*p* < 0.0001) [33].

Therefore, a first-line cBTKi seems to be the most appropriate choice for patients with unmutated IGHV, especially the elderly/unfit. For younger patients with unmutated IGHV, as will be shown below, the long estimated PFS obtained with the new fixed duration therapies make this option more attractive. When choosing between the different first-line cBTKis, given that no comparative trial is available to date but only indirect comparisons, we prefer to use acalabrutinib or zanubrutinib, given their similar efficacy and lower toxicity compared to ibrutinib. The main results of the first-line trials are listed in Table 2.

#### 2.2.2. Venetoclax–Obinutuzumab

First-line fixed-duration venetoclax–obinutuzumab was compared with chlorambucil–obinutuzumab in the CLL14 trial in 432 elderly and/or unfit CLL patients (median age 72 years, median cumulative illness rating scale [CIRS] score 8, and del17p in 8%). After a median follow-up of 76.4 months, the estimated 6-year PFS was longer with venetoclax–obinutuzumab than with chlorambucil–obinutuzumab (53.1% vs. 21.7%, *p* < 0.0001); the estimated 6-year OS showed a difference in favor of venetoclax–obinutuzumab, although statistical significance was not yet reached (78.7% vs. 69.2%, *p* = 0.052) [27]. Of note, contrary to what was observed with BTKis, unmutated IGHV remains a negative predictor for venetoclax–obinutuzumab treatment (median PFS not reached vs. 64.8 months for mutated and unmutated IGHV, respectively, *p* < 0.001) [27]. The CLL13 trial randomized 1:1:1 a total of 926 CLL patients to receive first-line chemoimmunotherapy, venetoclax-rituximab, venetoclax–obinutuzumab or venetoclax–obinutuzumab–ibrutinib. Measurable undetectable residual disease (uMRD) and PFS were superior for venetoclax–obinutuzumab +/− ibrutinib compared with venetoclax–rituximab and chemoimmunotherapy, confirming obinutuzumab as the best anti-CD20 to be combined with venetoclax, and venetoclax-obinutuzumab combinations being superior to chemoimmunotherapy also in younger patients [34]. Of note, serious infections were more frequent in the chemoimmunotherapy group and the triple combination group than in the other two groups, so probably some of the benefits of triple therapy were neutralized by the need for dose reduction and early treatment discontinuation due to adverse events. The main results of the first-line trials are listed in Table 2.

#### 2.2.3. Ibrutinib–Venetoclax

BTK inhibition targets adhesion and homing to lymph nodes, so ibrutinib mobilizes CLL cells out of protective lymphoid niches and inhibits proliferation [43]. Venetoclax sensitivity differs between compartments due to a different antiapoptotic profile of CLL cells in lymph nodes as opposed to peripheral blood [44], conferring a lower PFS in patients with voluminous lymphadenomegaly [27,45]. On this basis, the combination of ibrutinib and venetoclax (IV) was tested in previously untreated CLL.

A phase two study of the combination of ibrutinib and venetoclax enrolled 80 CLL patients with unfavorable prognostic markers or who were older (≥65 years). With that combination, uMRD in bone marrow was achieved in 56% of patients after 12 cycles of treatment and there was a 3-year PFS of 93% [35]. In the phase two CAPTIVATE trial, 159 patients (median age 60 years) received three cycles of ibrutinib monotherapy to reduce the risk of tumor lysis syndrome, followed by ibrutinib–venetoclax combination therapy for twelve cycles, stopping treatment after fifteen cycles regardless of MRD status (fixed-duration cohort). The results obtained were superimposable with 60% of patients achieving uMRD in bone marrow and 4.5-year PFS and OS of 70% and 97%, respectively [36,37]. In patients with del(17p) and/or mutated TP53 (27/159, 17% of patients), ORR and CR were superimposable compared to the treated population (96% vs. 96%, and 56% vs. 55%, respectively); slightly lower was the estimated 24-month PFS, which was 96% (95% CI, 91–98) in patients without del(17p), and 84% (95% CI, 63–94) in patients with del(17p)/mutated TP53. In the same study, 154 patients underwent secondary randomization after 15 cycles according to MRD status (MRD cohort), and at the end of the 15-cycle treatment 68% of patients achieved uMRD [38].

On this basis, in the phase three GLOW trial, 211 elderly/unfit patients (age >65 years and/or CIRS >6 and/or creatinine clearance <70 mL/min) were randomized 1:1 to receive either chlorambucil–obinutuzumab or ibrutinib–venetoclax (three cycles ibrutinib “lead in”, followed by 12 months of combination). After 57 months of median follow-up, the estimated PFS at 54 months was 66.5% vs. 319.5% for venetoclax–ibrutinib and chlorambucil–obinutuzumab, respectively (HR 0.256; *p* < 0.0001). Ibrutinib–venetoclax achieved deeper uMRD (<10^−5^) rates in bone marrow and peripheral blood, respectively, in 40.6% and 43.4% of patients [39,40,41]. Patients in the GLOW trial were older (median age 71 years) and/or with comorbidities, and apparently experienced more AEs, especially cardiovascular AEs, than observed in phase two trials, with grade 3/4 diarrhea 10.4%, atrial fibrillation 6.6%, infections 15.1%, and hypertension 7.5% [40,41].

Finally, in the FLAIR trial [42], the association of ibrutinib and venetoclax was administered for a duration determined by measurable residual disease for up to 6 years of therapy. A total of 260 patients were randomized to IV, while 263 patients to FCR. After a median follow-up of 43.7 months, the estimated 3-year PFS was 97.2% with IV and 76.8% with FCR (HR 0.13, 95% CI 0.07–0.24). PFS was superior in IGHV-unmutated patients (HR 0.07, 95% CI 0.02–0.19) but not in mutated patients (HR 0.54, 95% CI 0.21–1.38). The 3-year OS was 98% with IV and 93% with FCR (HR 0.31, 95% CI 0.15–0.67). The risk of infection was similar in the IV and FCR groups (21.5% vs. 17.1%, respectively); the percentage of cardiac serious adverse events was 10.7% in the IV group and 0.4% in the FCR group [42].

Although comparisons between different studies are not possible, mindful of the burden of possible toxicities that emerged from all trials, especially from the cardiovascular point of view, we suggest that the combination of ibrutinib and venetoclax should be preferred in young first-line patients, particularly with unmutated IGHV genes, thus replacing the use of venetoclax–obinutuzumab in this population, and should be considered also in del(17p)/TP53 mutated young patients. For the treatment at relapse after IV, retreatments with either ibrutinib or venetoclax–rituximab are both feasible options with promising results, as will be discussed later in this review. This combination has been approved for CLL by the FDA, the EMA, and the AIFA. The main results of the first-line trials are listed in Table 2.

#### 2.2.4. Future Perspectives

Failure to achieve uMRD at the end of treatment has a negative prognostic impact in CLL and is associated with a lower PFS after VO [27]. For this reason, frontline new therapeutic regimens including BTKi and venetoclax +/− CD20 are being investigated, often combined with MRD-guided treatment discontinuation, with the goal of inducing higher rates of uMRD and ultimately improving outcomes.

Two phase three trials are currently underway comparing the efficacy of the doublet combination of BTKi and venetoclax versus venetoclax–obinutuzumab, the CLL17 study for ibrutinib vs. VO vs. IV (NCT04608318), and the MAJIC study for acalabrutinib plus venetoclax vs. VO (NCT05057494).

The triplets, obtained with the addition of BTKi to venetoclax and anti-CD20, resulted in uMRD rates in bone marrow between 66% and 89% when combining VO with ibrutinib, acalabrutinib, and zanubrutinib [46,47,48,49]. A high rate of BM-uMRD was also shown in patients with TP53 disruption, both in the phase three CLL2-GIVe trial, in which 41 patients with 17p deletion were treated with IV plus obinutuzumab [IVO] (66% BM-uMRD, 3y PFS 80%) [47] and from subgroup analyses of a phase two trial, in which 10 patients with TP53 disruption were treated with acalabrutinib, venetoclax, and obinutuzumab [AVO] (92% BM-uMRD) [48].

The combination of ibrutinib with VO is currently the only triplet currently compared with the standard of care, and it did not substantially improve the depth of treatment response while increasing toxicity. In the CLL13 study, the difference in BM-uMRD between the two VO and IVO arms was minimal (73% vs. 78%) and the 3-year PFS was similar (88% vs. 91%) [34]. Nevertheless, the IVO arm of the study had more adverse events than VO, especially cardiovascular toxicity (atrial fibrillation 7.8% of patients). In phase two studies with selective BTKi however, cardiovascular toxicity was less; notably, 3% of patients treated with AVO [48] or zanubrutinib plus VO [49] experienced atrial fibrillation. Regarding IVO, two more randomized phase three trials with identical treatment arms (ibrutinib and obinutuzumab, IV, IVO) are underway and will test the efficacy and safety of ibrutinib-based combinations in elderly (Alliance A041720, NCT03737981) and young (ECOG E9161, NCT03701282) CLL patients.

Two phase three trials are currently investigating triplet combinations including acalabrutinib instead of ibrutinib: one comparing acalabrutinib–venetoclax vs. AVO vs. FCR (NCT03836261), and the other (CLL16, NCT05197192) comparing VO vs. AVO in patients with high-risk CLL, harboring TP53 aberration or complex karyotype.

Ongoing clinical trials will need to be completed in order to obtain more safety data and a longer follow-up and thus be able to adequately tailor the treatment for each patient. In the future, the use of fixed-duration combination therapies with two (doublets) or three (triplets) drugs will also be conditioned by the possibility of retreating patients with the same class of drugs in the case of relapse with need for treatment.

## 3. Previously Treated CLL

Patients with progressive disease must still meet iwCLL 2018 treatment criteria to have an indication to start a new treatment or switch from one treatment to another. Choosing the most appropriate therapy in patients with relapsed/refractory CLL requires not only, as with first-line CLL therapy, an assessment of the patient’s comorbidities, frailty and TP53 mutational pattern, but also further consideration of previously received therapies and outcomes. Chemoimmunotherapy is no longer used to treat patients with relapsed/refractory CLL at the current time. The main results of the studies in patients with relapsed CLL are listed in Table 3. Outside clinical trials, our current approach to relapsed/refractory patients who meet the 2018 iwCLL criteria for therapy is shown in Figure 2.

In the case of progression during continuous treatment, patients should be screened for BTK, phospholipase C-γ2 (PLCG2), or BCL2 gene mutations. The most frequent BTK mutation emerging in CLL patients on ibrutinib and acalabrutinib is in their binding site (C481 residue), causing an aminoacidic substitution (most commonly C481S) and resulting in a binding at a lower affinity [62,63,64]. Gain-of-function mutations in PLCG2 result in constitutive activation irrespective of BTK activation and thus continuous downstream BCR pathway signaling [62,64]. Recently, a study evaluating the incidence of mutations in five clinical trials showed that patients without PD on frontline cBTKis rarely develop mutations in BTK (3%), PLCG2 (2%), or both genes (1%), compared to relapsed/refractory patients (30%, 7%, and 5%, respectively). These data were also confirmed in patients evaluated at PD, with a lower incidence of mutations in previously untreated compared to relapsed/refractory patients, both in BTK (25% vs. 49%) and PLCG2 (8% vs. 13%) [65].

Different mutations of the BCL2 gene can confer resistance by reducing or inhibiting the binding affinity of venetoclax for the Bcl2 protein: G101V modulates the BH3 binding domain reducing venetoclax activity in vitro [66,67,68], D103Y directly disrupts the BH3 binding P4 pocket, and various other mutational patterns lead to the overexpression of the pro-survival proteins BCL-XL and MCL1 [66,67,69,70]. BCL2 mutations have been detected in relapsed/refractory patients, especially those treated with continuous venetoclax treatment. In a recent study, screening for BCL2 G101V and D103Y in 67 relapsed/refractory CLL patients on venetoclax revealed their presence in 10.4% and 11.9% of the cases, respectively, with four patients harboring both resistance mutations. Out of 11 patients carrying BCL2 G101V and/or D103Y, 10 experienced a relapse, accounting for 43.5% of disease progressions. All of these mutations were detected in patients on continuous venetoclax, none were observed during or after fixed-duration treatment [71]. Similarly, currently no acquired mutations in BCL2, BIM, BAX, BCL-XL, or MCL1 conferring resistance have been detected in the CLL14 trial after the first-line fixed-duration combination treatment with venetoclax–obinutuzumab [72]. Based on the evidence to date, the search for mutations that confer resistance is a clinical concern especially after the second relapse; anyway, more data are needed to recommend screening for mutations only in that context, and it is therefore appropriate in all patients progressing after a treatment with inhibitors.

### 3.1. BTKi- and BCL2i-Naïve

#### 3.1.1. Covalent BTKi

Both BTKis and BCL2is have robust efficacy and safety data in patients with CLL relapse after chemoimmunotherapy. Decision-making on the choice of treatment between BTKi and venetoclax requires a consideration of comorbidities, but also takes into account possible logistical difficulties and patient preference. Certainly, however, the considerations performed for the first line regarding patients with unmutated IGHV genes and/or TP53-disrupted genes remain valid. A critical element of the discussion is the evaluation of drug toxicity profiles, and it is of paramount importance to educate the patient on the optimal management of adverse events that might occur.

Ibrutinib as a continuous therapy was compared with ofatumumab at fixed duration for 24 weeks in the phase three RESONATE trial in 391 patients with relapsed CLL. In the last update (median follow-up 65.3 months), the median PFS was 44 months with ibrutinib and 8 months with ofatumumab, respectively (*p* < 0.001); the crossover rate to ibrutinib was 68%, so no difference in OS was observed [50]. In the phase three ASCEND trial, acalabrutinib as a continuous therapy was compared with the investigator’s choice of idelalisib (continuous) plus rituximab (eight infusions) or BR (six cycles) in 398 patients with previously treated CLL with a median of two prior therapies [51]. At a median follow-up of about 4 years, acalabrutinib showed superior outcomes in both PFS (62% vs. 19%) and OS (78% vs. 65%) [52]. The phase three ELEVATE-RR study then compared acalabrutinib and ibrutinib in relapsed/refractory CLL. The study randomized 533 patients with R/R CLL (45% harboring del17p) to receive acalabrutinib or ibrutinib and, at a median follow-up of 41 months, demonstrated the noninferiority of acalabrutinib with a median PFS of 38 months in both treatment arms. When comparing safety, acalabrutinib showed fewer discontinuations due to adverse events (21.3% vs. 14.7%) and a lower incidence of adverse events of interest, including bleeding (38% vs. 51%), hypertension (8.6% vs. 22.8%), atrial fibrillation (9% vs. 15.6%), and arthralgia (15.8% vs. 22.8%) compared with ibrutinib [53].

Zanubrutinib was compared with ibrutinib in 652 patients with R/R CLL in the phase three ALPINE trial, demonstrating superiority in both PFS (3-year PFS 64.9% vs. 54.8%, HR 0.68, *p* = 0.0011) and safety, with a lower incidence of cardiac events in the zanubrutinib arm (21.3% vs. 29.6%) and lower incidence of atrial fibrillation/flutter (5.2% vs. 13.3%). Of note, among patients with TP53 disruption (23% of patients in each treatment arm), those who received zanubrutinib had longer PFS than those who received ibrutinib (3-year PFS 58.6% vs. 41.3%, HR 0.52; 95% CI, 0.33–0.83) [54,55,56].

There are no trials comparing acalabrutinib and zanubrutinib directly, but indirect comparisons have been made using meta-analysis [73] and matching-adjusted indirect comparison (MAIC) [74].

Using unanchored MAIC on patients enrolled in the ASCEND trial, 99 patients treated with acalabrutinib were matched with patients treated with zanubrutinib in the ALPINE trial, weighting individual patient data to reduce differences in potentially confounding variables. The study showed a non-inferiority of acalabrutinib compared to zanubrutinib (2-year PFS 76% vs. 78%, HR 0.90, 95% CI 0.60–1.36), and lower incidence with acalabrutinib of having a serious AE (HR 0.61, 95% CI 0.39–0.97), hypertension (any grade: HR 0.18, 95% CI 0.09–0.37; grade ≥ 3: HR 0.22, 95% CI 0.09–0.54), any grade hemorrhage (HR 0.54, 95% CI 0.34–0.87), or an AE leading to dose reduction (HR 0.30, 95% CI 0.14–0.67) [74]. Nevertheless, limitations of MAIC analyses mean the results should be viewed as hypothesis-generating. Concerning safety, a meta-analysis including 61 trials, 6959 patients and 68 treatment arms containing ibrutinib, acalabrutinib, or zanubrutinib, has recently outlined more specific safety profiles for each drug, showing an improved AE profile with acalabrutinib and zanubrutinib compared to ibrutinib, but a similar average incidence of adverse events between the two drugs. The most frequent grade ≥ 3 AEs with zanubrutinib compared to acalabrutinib were neutropenia (HR = 1.43), cellulitis (HR = 6.6), and upper respiratory tract infection (HR = 2.09). In contrast, the most frequent grade ≥ 3 AEs with acalabrutinib compared to zanubrutinib were anemia (HR = 0.58), infection (HR = 0.76) and rash (HR = 0.03) [73]. The limitations of the meta-analyses were that the populations were not homogeneous across the trial, or for tumor type (CLL trials were 53%).

Therefore, when choosing a BTKi for a relapsed/refractory CLL, we consider more appropriate treatments with acalabrutinib or zanubrutinib given the similar efficacy and lower toxicity compared to ibrutinib. Based on the results of the ALPINE trial [55], in TP53-disrupted CLL patients we suggest favoring the use of zanubrutinib if available. The considerations made in the frontline setting for adding anti-CD20 monoclonal antibodies in combination also stand for the relapsed/refractory setting.

The main results of the trials for relapsed/refractory patients are listed in Table 3.

#### 3.1.2. Venetoclax +/− Anti-CD20

In the phase three study MURANO, venetoclax (for 2 years) combined with rituximab (6 months) showed superior efficacy compared with BR in 389 patients with relapsed CLL (27% with del17p, 2.6% exposed to B-cell receptor inhibitor) [59]. At a median follow-up of 59 months, the median PFS was 53.6 months with venetoclax plus rituximab (vs. 17 months with BR), and 5-year OS was 82% (vs. 62% with BR, *p* < 0.0001). The 5-year PFS rate was 38% among all patients receiving venetoclax plus rituximab but was lower among those with TP53 aberrations (27%), unmutated IGHV (29%), and genomic complexity (18%, defined by the presence of three or more copy number alterations) [60]. In the last update at a median follow up of 86.8 months, 7-year PFS rates were 23.0% with venetoclax–rituximab, while no patients treated with BR remained progression-free at this time point; 7-year OS rates were 69.6% vs. 51.0% (HR 0.53, *p* < 0.0001); median time to next treatment with venetoclax–rituximab was 63 months vs. 24 months with BR (HR 0.30, *p* < 0.0001) [61].

Venetoclax as continuous monotherapy was evaluated in a phase two study of 158 patients with del17p and showed a median PFS of 28 months and a median OS of 62 months [57,58]. The continuation of venetoclax beyond 2 years after combination with anti-CD20 may be considered in patients with del17p and/or TP53 mutation or in those in whom uMRD was not achieved at the end of treatment. However, in the MURANO study, patients with TP53 mutation relapsed rapidly, so the continuative treatment with venetoclax may not be effective. Anyway, when choosing BCL2i therapy for a relapsed/refractory BTKi-naive patient, we usually suggest combination with anti-CD20 for low-risk patients (IGHV mutated without TP53/del17p mutations); other patients could be treated with continuous venetoclax if not eligible for BTKi.

In the future, when all the data will be available, it will be necessary to combine them to determine which treatment is most effective in this context, and with this in mind, it will be crucial to investigate the mutations that confer resistance to each drug (e.g., BCL2 mutations [66,67,69,70], BTK mutations [62,63,64,75,76,77], PLCG2 mutations [62,78,79,80,81], etc.) to take full advantage of the potential of each treatment before switching to other molecules. The main results of the trials for relapsed/refractory patients are listed in Table 3.

### 3.2. Previous BTKi, Venetoclax Naïve

#### 3.2.1. Previous Treatment Stopped due to Toxicity (BTKi-Exposed)

The reason for the discontinuation of BTKi therapy is crucial when choosing the next treatment for a patient previously exposed to a BTKi. Indeed, suspensions due to toxicity account for approximately 50% of discontinuations [82,83,84]. Upon the resumption of treatment, dose reduction may be an option, and a further option is treatment with an alternative covalent BTKi in order to take full advantage of the potential efficacy of this class of drugs. A phase two study showed that acalabrutinib in patients intolerant to ibrutinib (persistent grade 3/4 or persistent/recurrent grade 2 adverse event despite dose modifications/interruption) was well tolerated, with 40% of patients experiencing the ibrutinib-intolerance adverse event, 67% of which were of lower grade on acalabrutinib than on previous ibrutinib [85]. Similarly, in a phase two study exploring zanubrutinib safety in covalent-BTKi-intolerant patients, both ibrutinib and acalabrutinib intolerance events recurred in 30% and 25% of patients intolerant to ibrutinib and acalabrutinib, respectively. When patients experiencing grade 3 adverse events while on ibrutinib or acalabrutinib were retreated with zanubrutinib, in 92% and 25% of patients the side effects recurred at a lower severity, respectively [86].

The choice to rechallenge a covalent BTKi after an adverse event must take into consideration the type of toxicity and its severity: major/recurrent cardiologic toxicities and major bleeding configure a scenario in which a rechallenge with the same drug class is less appropriate compared with diarrhea, rash, arthralgias, or even hypertension. It is important to remember that following prolonged treatment discontinuation, retreatment should only be considered when the patient again meets iwCLL 2018 treatment criteria, as the response achieved until discontinuation for toxicity may be maintained for years (median PFS after discontinuation for toxicity 25 months in the E1912 trial [30]).

#### 3.2.2. Previous Treatment Stopped due to Progression (BTKi-Refractory)

Patients who experience disease progression during BTKi therapy require a different therapeutic approach. Since covalent BTKis share the cysteine 481 binding site, the mechanisms causing resistance are similar, and therefore, switching to another drug of the same class will not provide any clinical benefit [62,63,64,75,76,77,78,79,80,81]. We suggest testing mutations of BTK and PLCG2 in patients progressing during BTKi therapy.

Venetoclax-based treatment represents the standard of care for BTKi-refractory and venetoclax-naive patients. Not many prospective data are available in this setting. A phase two study showed that patients previously treated with ibrutinib (55% with disease progression with ibrutinib, del17p 44%, and a median of four prior lines) treated with venetoclax monotherapy achieved an ORR of 65% and a median PFS of 24.7 months [87]. Since the majority of data regarding treatment in BTKi-refractory patients come from studies with venetoclax monotherapy, it should be considered whether to treat such patients with continuous therapy, even beyond 2 years if given with an anti-CD20.

Data from retrospective studies confirm venetoclax-based treatments as a highly effective option in BTKi-resistant patients [88,89]. In a retrospective study, 144 patients undergoing disease progression on ibrutinib were evaluated, showing a median OS of 29.8 months with subsequent venetoclax-based treatment compared with 9.1 months with other treatments (chemoimmunotherapy, PI3K inhibitors, and anti-CD20 monotherapy) [90].

Pirtobrutinib and nemtabrutinib are two noncovalent BTKis that have shown promising efficacy in BTKi-refractory patients, regardless of the presence or absence of C481S resistance mutations [91,92]. In addition, pirtobrutinib showed an excellent safety profile, with only 1% of patients discontinuing treatment due to a drug-related adverse event. Neither of these two drugs is currently available for CLL outside clinical trials, but upon approval they will represent a viable option in patients who have progressed while undergoing covalent BTKi therapy.

### 3.3. Previous Venetoclax, BTKi Naïve

#### Previous Treatment Stopped due to Toxicity or Treatment Completion (Venetoclax-Exposed) or due to Progression (Venetoclax-Refractory)

As for BTKi, the choice of treatment after venetoclax must take into consideration the response achieved with venetoclax and the reason for discontinuation: discontinuation due to toxicity, progression during treatment, or progression after terminating fixed-duration therapy.

The best choice for patients with unacceptable toxicity or disease progression on venetoclax is covalent BTKi. Again, as in previous cases, patients should only be retreated if iwCLL 2018 treatment criteria are present.

A large multicenter retrospective study evaluated the response to post venetoclax therapies in relapsed/refractory CLL. Of 74 patients treated with BTKi after venetoclax, 44 BTKi-naive patients achieved an ORR of 84% and a median PFS of 32 months with BTKi after venetoclax [93]. These results were lower in BTKi-exposed and BTKi-refractory patients (see next paragraph). In another retrospective study of 23 venetoclax-refractory patients, similar results were obtained on subsequent BTKi therapy, with a median PFS of 34 months and an ORR of 90% [94]. In this study, authors also highlighted that the duration and depth of responses to venetoclax have a negative impact on PFS after BTKi initiation (CR/uMRD vs. non-CR/uMRD HR 0.15, *p* = 0.029; months on venetoclax ≥ 24 vs. <24 months HR 0.31, *p* = 0.044) [94].

In the CLL14 trial, 61.7% of the 67 relapsed patients who required subsequent therapy after venetoclax–obinutuzumab received BTKi, but results for BTKi efficacy are not yet available [27]. In the MURANO trial, 27% (n = 18) of relapsed patients who required subsequent therapy after venetoclax–rituximab received BTKi, obtaining an ORR of 100% [61]. Regarding venetoclax restart after a fixed-duration venetoclax-based treatment, the depth and the duration of the response should be carefully evaluated. Although we have no prospective data on the actual duration of response to perform a rechallenge with venetoclax after discontinuation, it is likely that progressing patients will benefit from a second venetoclax-based treatment if the duration of response has exceeded at least one year. In the currently still ongoing crossover/retreatment substudy of the MURANO trial, of 34 relapsed patients, 25 were rechallenged with venetoclax–rituximab after a median time from the end of previous treatment of 2.3 years (88% IGHV unmutated, 32% TP53 disrupted, and 32% genomic complexities >5) [61]. After a median follow up of 33.4 months, median PFS was 23.3 months, with a best ORR of 72%, indicating that venetoclax–rituximab is a viable option for pretreated patients. A multicenter retrospective study of 46 patients who underwent venetoclax-based treatments (monotherapy, + anti-CD20, + BTKi) and were then retreated again with a venetoclax-based approach (median time to retreatment 16 months), showed an ORR of 79.5% and a median PFS of 25 months. Of note, in the subgroup of patients with BTKi exposure before the first treatment with venetoclax (n = 18), at the second treatment the ORR was 56.3% and the median PFS was 15 months [95].

A phase two study is also underway to test retreatment with venetoclax–obinutuzumab (ReVenG NCT04895436). Currently, the evaluation of BCL2 mutations is not part of the decision-making algorithm for retreatment with venetoclax after previous fixed-duration treatment, but we still recommend performing it if possible, given the still limited amount of data available in this setting.

### 3.4. Previous BTKi and Venetoclax (Double-Exposed and Double-Refractory)

Patients who have received covalent BTKi and venetoclax (“double-exposed”) need careful evaluation to receive individualized treatment based on clinical history. Retreatment with the same class should be considered in the case of discontinuation for reasons other than progression, in accordance with what is discussed in the previous sections. In the case of relapse after fixed-duration therapy with both (venetoclax + cBTKi) in patients who achieve a response and relapse after a discrete period of time, restarts of either venetoclax [96] or ibrutinib [37] are currently under investigation in clinical trials, with promising results. In the CAPTIVATE trial, to date, 22 patients with progressive disease after completion of Ibr + Ven at fixed duration have been retreated with ibrutinib. Of 21 evaluable patients, ORR was 86%, with a median duration of retreatment of 17 months (range 0–45) [37]. It should be noted that no BTK, PLCG2, or BCL2 resistance mutations were identified in patients who experienced progression, suggesting that retreatment with one or both agents is a viable strategy [36]. In the IMPROVE trial, 13 patients were retreated with venetoclax (seven of whom previously received IV therapy) at a median of 32 months (12–41) after the end of initial therapy. The progression-free survival of retreatment with venetoclax was 14 months (0–23) [96].

Patients with disease progression undergoing treatment with both drug classes (“double-refractory”) represent an unmet clinical need, as no effective treatment options are available. To date, the time to subsequent line failure or death for patients with CLL progression in this setting was 3.9 months [97], and the median OS was only 8 months [98]. Therefore, it becomes critical to enroll such patients in clinical trials, especially patients with early relapse. Small retrospective studies have suggested that retreatment with cBTKi in combination with venetoclax may provide additional benefit even in double-refractory patients [99,100], although it is mostly to be considered a bridge treatment to other therapies.

### 3.5. Future Perspectives

The emerging treatment effort in patients with relapsed/refractory CLL is currently to overcome BTK, PLCG2, and BCL2 mutation-dependent resistance. In this context, the treatments that have proven to be effective and for which we have more data available are non-covalent BTKis (pirtobrutinib and nemtabrutinib) and chimeric antigen receptor T-cell therapy (CAR-T), which will be discussed in detail below.

Other promising treatments include bispecific antibodies, particularly epcoritamab in the recently expanded phase 1b/2 EPCORE-CLL1 trial (NCT04623541), BTK-degraders (NCT04830137, NCT05131022, NCT05780034, and NCT05006716), pyruvate kinase C-β (PKC-β) inhibitors (NCT03492125), and dual irreversible/reversible BTKis (NCT04775745).

The main results of the trials exploring new therapies for relapsed/refractory patients are listed in Table 4.

#### 3.5.1. Non-Covalent BTKi

In the phase 1b/2 BRUIN study, pirtobrutinib achieved an ORR of 73.3% (95% CI, 67.3 to 78.7), 82.2% when partial response with lymphocytosis was included (95% CI, 76.8 to 86.7), with responses in all patient cohorts, including those with mutations conferring resistance to cBTKis, with the exception of the 18 patients with PLCG2 mutation which obtained an ORR of 56% (95% CI, 31 to 79). In the double exposed/refractory group, ORR was 70.0% (95% CI, 60.0 to 78.8), 79.0% (95% CI, 69.7 to 86.5) when partial response with lymphocytosis was included. At a median follow-up of 19.4 months, median PFS was 19.6 months (95% CI, 16.9 to 22.1). Median PFS was 22.1 months (95% CI, 19.6 to 27.4) in cBTKi-exposed patients, and 16.8 months (95% CI, 13.2 to 18.7) in double-exposed/refractory patients. Similar estimates of PFS were observed regardless of BTK C481, del(17p)/TP53, and IGHV mutation status. At a median follow-up of 22.6 months, the 1-year OS of cBTKi-exposed patients was 86% (95% CI, 81.0 to 89.8).

Overall, the most common adverse events were infections (71.0%, grade ≥ 3 28.1%), bleeding (in 42.6%, grade ≥ 3 2.2%), and neutropenia (in 32.5%, grade ≥ 3 26.8%). No cases of sudden cardiac death, drug-related case of ventricular fibrillation, or ventricular tachycardia were observed [91].

Several phase three trials are currently underway exploring the use of pirtobrutinib as a monotherapy (NCT04666038, NCT05023980, NCT05317936, and NCT05254743) and in combination (NCT04965493, NCT05677919, and NCT05536349), both for untreated and relapsed/refractory patients.

The reported follow-up with nemtabrutinib is relatively shorter, reporting an ORR of 58% and a median duration of response among responders of 24.4 months (95% CI, 13.9—not evaluable), showing durable responses even in a population previously treated with BTKis [92].

#### 3.5.2. CAR-T

The efficacy of all of the CD19-targeted CAR-T cell therapies currently available has been evaluated in CLL, but the most consistent data available are those concerning lisocabtagene maraleucel (liso-cel).

In the TRANSCEND CLL 004 trial, 117 patients received liso-cel, achieving an ORR of 82% (CR 45%), with peripheral blood uMRD rates of 75% and bone marrow uMRD rates of 65%. The subgroup of 49 double-refractory patients, 86% of whom had high-risk cytogenetics, had an ORR of 42%, but with a median PFS among responders of 35.3 months. In exploratory analyses, median PFS was 26.2 months in patients with uMRD in peripheral blood and 2.8 months in those with detectable MRD. Grade 3 CRS occurred in 8.5% of cases and grade 3 neurological events in 17.9% of cases [101].

In this study, a cohort of patients received liso-cel in combination with ibrutinib, achieving an ORR of 95% (CR 47%), with uMRD rates of 89% in peripheral blood and 79% in bone marrow [101]. To date, two other studies have reported an encouraging efficacy of the concurrent administration of ibrutinib and CD19-targeted CAR-T cell therapy. In a phase I/II pilot study, in 19 patients 4-week ORR was 83% (bone marrow uMRD 61%), resulting in 1 yr OS and PFS of 86% and 59%, respectively [102]. In the other phase II study, anti-CD19 humanized binding domain CAR-T cells were administered in combination with ibrutinib in 19 patients not in CR after ≥6 months of ibrutinib, obtaining a CR in 44% of patients; 4y PFS among responders was 70% [103]. Complete responses were also obtained with CAR-NK cells (NCT05410041) in patients with heavily pretreated disease and exposed to novel agents [104]. However, more consistent data are needed before these therapies can be introduced into the treatment algorithm of patients with R/R CLL, so enrolling such patients in clinical trials is crucial.

### 3.6. Allogeneic Hematopoietic Stem Cell Transplantation

CLL remains incurable despite the efficacy of the targeted agents described above, especially in patients at high cytogenetic risk. In the largest retrospective study of allogeneic HSCT in 65 patients with CLL who had received at least one targeted agent, 24-month PFS and OS rates of 63% and 81%, respectively, were reported [105]. The incidence of recurrence was 27%, non-relapse mortality was 13%, and grade III–IV acute graft versus host disease at 100 days was 24%. The only predictor of worse PFS was hematopoietic cell transplantation-specific comorbidity index (HCT-CI) with an HR of 3.3 (95% CI 1.1–9.9, *p* = 0.035). Not independently associated with PFS were cytogenetic risk, previous exposure to targeted agents and/or chemoimmunotherapy, the number of targeted agents taken, complete or partial remission, transplant characteristics, or ibrutinib versus venetoclax as the line of therapy immediately prior to transplant [105].

Therefore, from our perspective allogeneic HSCT should be considered in patients with adequate clinical fitness and a low burden of medical comorbidities during the remission of a second targeted agent, especially in patients at high cytogenetic risk, given the poor long-term disease control in double-refractory patients despite new therapeutic approaches.

## 4. Autoimmune Cytopenias

Autoimmune cytopenias (AICs) (i.e., autoimmune hemolytic anemia (AIHA), immune thrombocytopenia (ITP) or, more rarely, pure red blood cell aplasia or autoimmune granulocytopenia), are a frequent complication of CLL, affecting 5–10% of patients [106,107,108]. Patients with AIC, according to the ESMO guidelines [12] and the International Workshop on CLL guidelines [4], should first receive direct treatment against the autoimmune manifestation. The first line involves the use of high-dose corticosteroids, which are effective especially in warm autoantibody forms. In patients relapsed or refractory to steroids, reasonable second-line treatment options could be rituximab alone or in combination with cyclophosphamide and dexamethasone [12], or the treatment of CLL. The use of targeted agents is progressively showing convincing efficacy in the treatment of these patients, although emergent AIC in the course of treatment should still be considered [108]. From our perspective, the already appropriate second-line approach is to treat the disease with targeted agents, especially in the steroid-refractory forms. An ongoing study is currently evaluating the use of BTKis as a frontline therapy for AICs in CLL patients (NCT05694312). For further discussion, we refer to specific reviews on this topic.

## 5. Conclusions

Recent years have witnessed a continuous evolution of the CLL treatment landscape due to significant advances in the efficacy and tolerability of new therapies. The best combinations and sequences of administration of such therapies will be the questions that will be answered in the coming years. Moreover, most of the available efficacy and safety data are derived from clinical trials and it will therefore also be necessary to integrate these results with real-life clinical practice in order to increasingly tailor treatments to patients. The challenge ahead will be the development and integration of new therapeutic agents and the ability to tailor treatment to improve the prognosis of double-refractory patients, currently still an unmet clinical need.

## Figures and Tables

**Figure 1 cancers-16-02011-f001:**
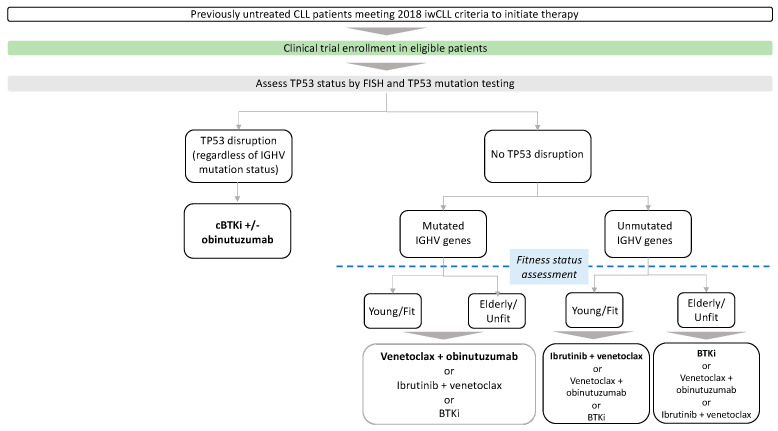
Frontline treatment. Preferred cBTKis are acalabrutinib and zanubrutinib in most patients. Preferred treatment is in bold. cBTKi: covalent Bruton tyrosine kinase inhibitor, IGHV immunoglobulin heavy chain gene.

**Figure 2 cancers-16-02011-f002:**
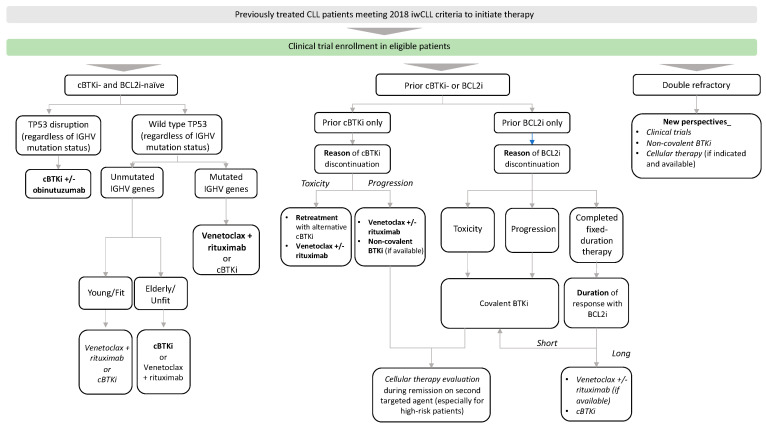
Treatment approach for previously treated CLL patients. Preferred cBTKis are acalabrutinib and zanubrutinib in most patients. Preferred treatment is in bold. Therapeutic alternatives with overlapping efficacy are in italics. cBTKi: covalent Bruton tyrosine kinase inhibitor, IGHV immunoglobulin heavy chain gene.

**Table 1 cancers-16-02011-t001:** International Workshop on CLL (iwCLL) indications for treatment [4].

Symptom	Indication for Treatment
Anemia	Anemia < 10 g/dL due to progressive marrow failure ^1^
Thrombocytopenia	Thrombocytopenia <100 × 10^9^/L due to progressive marrow failure ^1^
Lymphocyte count	Progressive ≥ 50% over a 2-month period, or lymphocyte doubling time < 6 months
(Hepato-)Splenomegaly	Massive (i.e., ≥6 cm below the left costal margin), progressive, or symptomatic
Lymphadenomegaly	Massive (i.e., ≥10 cm), progressive, or symptomatic
Constitutional symptoms	Disease-related symptoms ^2^
Autoimmune complications	Autoimmune complications poorly responsive to corticosteroids or current treatment
Extranodal involvement	Symptomatic or functional extranodal involvement (e.g., skin, kidney, lung, spine)

^1^ Hemoglobin and platelets counts may remain steadily below these cutoffs over a long period; this situation does not automatically require therapeutic intervention. ^2^ Unintentional weight loss ≥ 10% within the previous 6 months; significant fatigue (ECOG performance scale ≥ 2), fevers (38.0 °C) for ≥2 weeks without evidence of infection; night sweats for ≥1 month without evidence of infection.

**Table 2 cancers-16-02011-t002:** Frontline treatment trials.

Trial	Line of Therapy	Drug	Patients (n)	Median Age (Years)	TP53-Disrupted Patients	IGHV-Unmutated Patients	ORR	PFS	OS
RESONATE-2 (phase III) [28]	1L	Ibrutinib	136/269	73	11/124 (9%)	58/101 (57%)	92%	7y PFS 59%	7y OS 78%
Alliance A041202547 (phase III) [29]	1L	Ibrutinib	182/547	71	15/168 (9%)	77/122 (63%)	93%	4y PFS 76%	4y OS 85%
Alliance A041202547 (phase III) [29]	1L	Ibrutinib– Rituximab	182/547	71	20/168 (12%)	70/115 (61%)	94%	4y PFS 76%	4y OS 86%
E1912 (phase III) [30]	1L	Ibrutinib– Rituximab	354/529	58	27/299 (9%)	210/300 (75%)	95.8%	5y PFS 78%	5y OS 95%
FLAIR (phase III) [31]	1L	Ibrutinib– Rituximab	386/771	62	5/386 (1%)	194/386 (50%)	95.6%	4y PFS 85.6%	4y OS 92%
Ahn I, et al. (Phase II) [23]	1L	Ibrutinib	34/34	63	34/34 (100%)	21/34 (62%)	97%	6y PFS 61%	6y OS 79%
ELEVATE-TN (phase III) [9,32]	1L	Acalabrutinib	179/535	70	23/179 (13%)	119/174 (66.5%)	90%	6y PFS 62%	6y OS 76%
ELEVATE-TN (phase III) [9,32]	1L	Acalabrutinib– Obinutuzumab	179/535	70	25/179 (14%)	103/179 (57.5%)	96%	6y PFS 78%	6y OS 84%
SEQUOIA (phase III) [33]	1L	Zanubrutinib (arm A)	241/590	70	17/232 (7%)	125/234 (53%)	94.6%	2y PFS 85.5%	2y OS 94.3%
SEQUOIA (phase III) [25]	1L	Zanubrutinib (arm C)	109/109	70	109/109 (100%)	67/103 (65%)	94.5%	18-month PFS 88.6%	18-month OS 95.1%
CLL14 (phase III) [27]	1L	Venetoclax–Obinutuzumab	216/432	72	24/200 (12%)	121/200 (61%)	84.7%	6y PFS 53.1%	6y OS 78.7%
GAIA/CLL13 (phase III) [34]	1L	Venetoclax–Obinutuzumab	229/926	62	0/229 (0%)	130/228 (57%)	96.1%	3y PFS 87.7%	3y OS 96.3%
GAIA/CLL13 (phase III) [34]	1L	Venetoclax–Obinutuzumab– Ibrutinib	231/926	60	0/231 (0%)	123/231 (53.2%)	94.4%	3y PFS 90.5%	3y OS 95.3%
Jain N, et al. (phase II) [35]	1L	Venetoclax– Ibrutinib	80/80	65	Del17p 14/80 (18%) TP53m 11/79 (14%)	63/76 (83%)	100%	1y PFS 98%	1y OS 99%
CAPTIVATE (phase II) [36,37]	1L	Venetoclax– Ibrutinib (FD cohort)	159/159	60	27/159 (17%)	89/159 (56%)	96%	4.5y PFS 70%	4.5y OS 97%
CAPTIVATE (phase II) [38]	1L	Venetoclax– Ibrutinib (MRD cohort)	164/164	58	32/164 (20%)	99/164 (60%)	98%	NA	NA
GLOW (phase III) [39,40,41]	1L	Venetoclax– Ibrutinib	106/211	71	7/106 (6.6%)	55/106 (51.9%)	86.8%	4.5y PFS 66.5%	4.5y OS 84.5%
FLAIR (phase III) [42]	1L	Venetoclax– Ibrutinib	260/523	62	1/260 (0.4%)	123/260 (47.3%)	86.5%	3y PFS 97.2%	3y OS 98%

1L: frontline; ORR: overall response rate; PFS: progression-free survival; OS: overall survival; del17p: 17p deletion; TP53m: TP53 mutation; FD: fixed-duration; MRD: minimal residual disease; NA: not applicable.

**Table 3 cancers-16-02011-t003:** Treatments in patients with relapsed CLL.

Trial	Line of Therapy	Drug	Patients (n)	Median Age (Years)	Median Number of Previous Therapies	TP53-Disrupted Patients	IGHV-Unmutated Patients	ORR	PFS	OS
RESONATE (phase III) [50]	R/R	Ibrutinib	195/391	67	3 (1–12)	127/195 (65%)	98/134 (73%)	91%	5y PFS 40% Median PFS 44.1 mo	5y OS 51% Median OS 67.7 mo
ASCEND (phase III) [51,52]	R/R	Acalabrutinib	155/310	68	1 (1–8)	22/153 (14%)	118/154 (77%)	92%	42-month PFS 62% mPFS NR	42-month OS 78% mOS NR
ELEVATE-RR (phase III) [53]	R/R	Ibrutinib	265/533	65	2 (1–12)	135/265 (50.7%)	237/265 (89.4%)	77%	Median PFS 38.4 mo	3y OS 72.5% mOS NR
ELEVATE-RR (phase III) [53]	R/R	Acalabrutinib	268/533	66	2 (1–9)	136/268 (50.9%)	220/268 (82.1%)	81%	Median PFS 38.4 mo	3y OS 76.5% mOS NR
ALPINE (phase III) [54,55,56]	R/R	Ibrutinib	325/652	68	1 (1–12)	75/325 (23.1%)	239/325 (73.5%)	76%	3y PFS 54.8% mPFS NR	2y OS 81.5% mOS NR
ALPINE (phase III) [54,55,56]	R/R	Zanubrutinib	327/652	67	1 (1–6)	75/327 (22.9%)	239/327 (73.1%)	86%	3y PFS 64.9% mPFS NR	2y OS 85.3% mOS NR
Stilgenbauer et al. (phase II) [57]	R/R	Venetoclax	107/107	67	2 (1–4)	106/107 (99%)	30/37 (81%)	79%	1y PFS 72% mPFS NR	1y OS 86.7% mOS NR
Stilgenbauer et al. (phase II) [58]	1L + R/R	Venetoclax (expanded cohort)	158/158	67	2 (0–10)	157/158 (99.4%)	118/158 (75%)	77%	5y PFS 24% Median PFS 28.2 mo	5y OS 52% Median OS 62.5 mo
MURANO (phase III) [59,60,61]	R/R	Venetoclax– Rituximab	194/389	64.5	NA	65/171 (38%)	123/180 (68.3%)	92.3%	7y PFS 23% Median PFS 54.7 mo	7y OS 69.6% Median OS NR

1L: frontline; R/R: relapsed/refractory; ORR: overall response rate; (m)PFS: (median) progression-free survival; (m)OS: (median) overall survival; mo: months; NR: not reached; NA: not applicable.

**Table 4 cancers-16-02011-t004:** Future perspectives in patients with relapsed CLL.

Trial	Line of Therapy	Drug	Patients (n)	Median Age (Years)	Median Number of Previous Therapies	TP53-Disrupted Patients	IGHV-Unmutated Patients	ORR	PFS	OS
BRUIN (phase I/II) [91]	R/R	Pirtobrutinib (BTKi-exposed)	247/317	69	3 (1–11)	90/193 (46.6%)	168/198 (84.8%)	82.2%	Median PFS 19.6 mo	18-month OS 80.5%
BELLWAVE 001 (phase I/II) [92]	R/R	Nemtabrutinib	57/112	66	4 (1–18)	19/57 (33%)	30/57 (53%)	56%	Median PFS 26.3 mo	NA
TRANSCEND CLL 004 (phase I/II) [101]	R/R	Lisocabtagene maraleucel	117/117	65	5 (3–7)	54/117 (46%)	54/117 (46%)	43%	18-month PFS 46.9% Median PFS 17.9 mo	18-month OS 71% Median OS 43.2 mo

R/R: relapsed/refractory; ORR: overall response rate; (m)PFS: (median) progression-free survival; (m)OS: (median) overall survival; mo: months; NA: not applicable.

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
