# Peer review of "Treatment Sequencing in Chronic Lymphocytic Leukemia in 2024: Where We Are and Where We Are Headed"

_cancers, 2024, doi:10.3390/cancers16112011_

Round 1

Reviewer 1 Report

Comments and Suggestions for Authors

Fresa and colleagues present an up-to-date review about the treatment algorithms and sequencing in the context of CLL. From a clinical perspective, this is an expanding theme. The manuscript is well organized and properly discussed. I would suggest to discuss more about the possible use of venetoclax-ibrutinib combination in del17p/TP53 mutated patients. Thanks.

Author Response

Dear reviewer,

Thank you for your kind suggestion and the opportunity to revise our paper ‘Treatment sequencing in chronic lymphocytic leukemia in 2024: where we are and where we are headed’.

I have included the reviewer comment immediately after this letter and responded indicating exactly how we addressed each concern or problem and describing the changes we have made. The revisions have been approved by all authors and I have again been chosen as the corresponding author.

We hope the revised manuscript will better suit Cancers but are happy to consider further revisions, and we thank you for your continued interest in our research.

Sincerely,

Luca Laurenti, MD

MD, Department of Hematology

Fondazione Policlinico Universitario A. Gemelli IRCCS, Roma, Italia

Largo A. Gemelli, 8. Rome, Italy. 00168

Phone: +39 0630154180

Reviewer Comments, Author Responses and Manuscript Changes

Reviewer #1

Comment 1: ‘I would suggest to discuss more about the possible use of venetoclax-ibrutinib combination in del17p/TP53 mutated patients.’

Response: Thank you for the suggestion. We added a paragraph highlighting the data available in del17p/TP53 mutated patients (lines 208-212) and our suggestion (239-240).

Reviewer 2 Report

Comments and Suggestions for Authors

Fresa et al has submitted a manuscript for publication on treatment sequencing in CLL. This fast-moving area of research has dramatically changed the treatment of CLL patients. This review systematically goes over clinical trial data in CLL patients. This is timely but needs to be more than just a listing of clinical trail data. List below are my concerns.

1.      Under each section such as 2.1 Del17p/TP53, there needs to be a concluding sentence as to what the trail showed or did not show. This critical analysis will be provide the reader with a native as to where treatment stand today. As it reads now, the reader gets lost in all the clinical trail data.

2.      One of the major issues is drug resistance in CLL treatments. This is somewhat addressed in the relapse sections. There needs to be a broader discussion on the mechanisms of drug resistance and the rationale researchers have used to address it.

3.      Future perspectives was lacking direction. It was stating future clinical trails without context. What does researchers need to address to improve CLL therapy?

4.      Combinational therapies were a focus for many of the section but this issues with toxicity and side effects were only touched upon briefly. This is a major concern in CLL treatments and need to be more thoroughly discussed.

5.      CAR T cells needs to be a stand alone section as this could be used to overcome drug resistance.

6.      A discussion of the limitations of this clinical data should also be provided.

Comments on the Quality of English Language

Fine

Author Response

Dear reviewer,

Thank you for your kind suggestions and the opportunity to revise our paper ‘Treatment sequencing in chronic lymphocytic leukemia in 2024: where we are and where we are headed’. The suggestions offered have been immensely helpful.

I have included the reviewer comments immediately after this letter and responded to them individually, indicating exactly how we addressed each concern or problem and describing the changes we have made. The revisions have been approved by all authors and I have again been chosen as the corresponding author.

We hope the revised manuscript will better suit Cancers but are happy to consider further revisions, and we thank you for your continued interest in our research.

Sincerely,

Luca Laurenti, MD

MD, Department of Hematology

Fondazione Policlinico Universitario A. Gemelli IRCCS, Roma, Italia

Largo A. Gemelli, 8. Rome, Italy. 00168

Phone: +39 0630154180

Reviewer Comments, Author Responses and Manuscript Changes

Comment 1: ‘Under each section such as 2.1 Del17p/TP53, there needs to be a concluding sentence as to what the trail showed or did not show.’’

Response: Thank you for the kind suggestion. We added a concluding sentence to guide readers through the different paragraphs (lines 162-168, 235-240)

Comment 2: ‘There needs to be a broader discussion on the mechanisms of drug resistance and the rationale researchers have used to address it’.

Response: Thank you for the suggestion. We added a paragraph on emerging mutations (lines 300-330).

Comment 3: ‘Future perspectives was lacking direction. It was stating future clinical trails without context.’.

Response: Thank you for the suggestion. We rewrote the future perspectives in the relapsed/refractory setting and highlighted the rationale of intervention (lines 561-571) and added two paragraphs for the two options with more consistent data available (lines 573-621). We slightly modified the future perspectives in the frontline setting.

Comment 4: ‘Combinational therapies were a focus for many of the section but this issues with toxicity and side effects were only touched upon briefly. This is a major concern in CLL treatments and need to be more thoroughly discussed.’

Response: Thank you for the suggestion. Side effects for each trial were already reported, but we added some considerations in the treatment choice, highlighting the importance of safety (lines 166-168, 235-240, 278, 663).

Comment 5: ‘CAR T cells needs to be a stand alone section as this could be used to overcome drug resistance’

Response: Thank you for the revision. We created a stand-alone section as suggested (lines 598-621).

Comment 6: ‘A discussion of the limitations of this clinical data should also be provided.’

Response: Thank you for the suggestion. We have reported data exclusively from clinical trials, so the limitations of each are noted in each. Since the very fact of being data derived from clinical trials can be a limitation in real-life clinical practice, we have emphasised this as the main limitation in the choice of treatments (lines 663-665).

Round 2

Reviewer 2 Report

Comments and Suggestions for Authors

Authors addressed my concerns.